# CG-CLIP: Seeing Beyond Objects to Learn Corruption Robustness

## Abstract

Large pre-trained models like CLIP exhibit an object-centric bias, rendering them brittle for tasks like assessing robustness to common image corruptions. We hypothesize that this stems from low-information classification objectives that fail to learn robust, structural representations. To overcome this, we propose Corruption-Guided Finetuning (CGF), which regularizes the model by introducing a dense auxiliary task: predicting pixel-wise corruption maps. We introduce a principled three-stage curriculum learning strategy to effectively integrate this dense objective with the global classification task. Our model, CG-CLIP, improves out-of-distribution corruption detection accuracy on the challenging Caltech-256 benchmark from 88.0% to 97.45%, a $\sim 9$ point gain over a strong baseline, FLYP. This improvement is achieved with no additional inference overhead, as the auxiliary components are discarded after training. Our work shows that compelling models to solve richer, structurally-aware tasks is a direct path to more robust and generalizable AI.

## 1 Introduction

The advent of large vision-language models like CLIP (Radford et al., 2021) has provided a powerful foundation for modern machine perception (Awais et al., 2025). By learning from vast quantities of image-text data, these models acquire rich semantic representations. However, their training objective instills a fundamental object-centric bias: they are optimized to answer the question, "What is in this image?", by associating visual concepts with text. Consequently, they are often ill-equipped to assess an image's robustness to image corruptions, for example, answering, "Is this photograph free of artifacts?". This inherent focus on semantic content over artifact detection makes them surprisingly fragile when adapted to downstream tasks that depend on visual quality (Wang et al., 2023a). This limitation becomes starkly evident when applying a pre-trained CLIP model to the critical task of corruption classification. As direct proof, we visualized the features of clean and corrupted images using the frozen, off-the-shelf CLIP vision encoder. The result, shown in Figure 3 (a), is unequivocal: the embeddings for clean and corrupted images are completely intermingled, with no meaningful separation. This confirms that the base model, despite its powerful semantic knowledge, lacks an innate feature representation for visual quality, making it unsuitable for this task domain without significant adaptation.

In this work, we hypothesize that the limitation may stem more from the training objective than from the architecture itself. The objective of classification, mapping a high-dimensional input to a single label, provides relatively limited information, as the output is much simpler than the input. By rewarding the model for producing a single global output, an embedding or a label, these objectives encourage the network to discard fine-grained spatial details in favor of abstract semantic features. This incentivizes reliance on superficial correlations and prevents the model from building a true internal model of the visual world, which is why finetuning often destroys robustness to novel corruptions and domain shifts (Wortsman et al., 2022; Goyal et al., 2023; He et al., 2022; Chen et al., 2020). In contrast, a dense prediction task like saliency mapping asks the model to make a large number of structured predictions per image, offering a far richer and more guided learning signal.

To build truly robust systems, we must force them to predict more, to build a richer, more detailed model of the world they are observing. This is the core principle behind self-supervised learning, and we extend it here to a supervised context (Jaiswal et al., 2020; Jing & Tian, 2020). We propose

a training paradigm, Corruption-Guided Finetuning, where we augment the simple classification objective with a dense, high-information auxiliary task. We force the model not just to classify an image as corrupt, but to also predict a dense corruption map that localizes the artifacts at a pixel level.

This corruption mapping objective acts as a powerful regularizer. It is a "supervised pretext task" that cannot be solved by memorizing superficial statistics. To succeed, the model must learn to identify the "fingerprints" of corruption, effectively building an internal model of natural image statistics by learning what makes an image look authentic and free of artifacts. This forces the emergence of a feature space with superior corruption robustness and generalizability. However, integrating these two distinct objectives poses a challenge. The global, spatially-invariant classification task requires abstract features, while the local, pixel-wise corruption mapping task demands fine-grained spatial detail. To resolve this, we propose the "Adapt-Isolate-Tune" pipeline, a principled curriculum learning strategy (Bengio et al., 2009; Soviany et al., 2022). This methodology guides the model through a structured, stage-wise process of knowledge acquisition, ensuring the dense prediction task enriches the global feature representation rather than disrupting it. The code will be released publicly upon acceptance. Our contributions are:

- A training paradigm, Corruption-Guided Finetuning, where an auxiliary explainability task is used to significantly improve the OOD robustness (stylistic domain shift and corruption), of a primary classification task.

- We validate our central hypothesis that a dense auxiliary task can regularize a model for robustness to corruptions by achieving state-of-the-art results. Our method boosts OOD corruption detection accuracy on Caltech-256 from 88.00% to 97.45% with no additional computational overhead at inference time.

## 2  RELATED WORK

Our research is situated at the intersection of two critical areas in modern deep learning: the robust finetuning of large pre-trained models and the emerging paradigm of using model explanations as a direct supervisory signal.

### 2.1  THE CHALLENGE OF ROBUST FINETUNING FOR VLMS

Vision-language models (VLMs) like CLIP have achieved impressive zero-shot generalization by pre-training on large, noisy image-text datasets (Li et al., 2025), but their robustness is often fragile, standard finetuning can improve in-distribution accuracy while degrading out-of-distribution performance. Early method, such as Linear Probing, where a linear classifier is trained on frozen image embeddings using cross-entropy, aimed to balance feature preservation with accuracy. FLYP (Goyal et al., 2023) aligned finetuning with the original contrastive objective to prevent feature collapse. Unlike weight-space ensembling methods such as WiSE-FT (Wortsman et al., 2022), which enhance robustness by interpolating between the weights of a zero-shot and a fine-tuned model, our approach trains a single model to be intrinsically robust through a structured curriculum. More recent approaches incorporate auxiliary signals, particularly high-level semantics, to regularize training; for instance, Anchor-based robust finetuning (ARF) of vision-language models (Han et al., 2024) enriches class labels with captions and semantically similar examples to better preserve the original feature space. In contrast, our work targets covariate shifts caused by data corruption rather than semantic gaps (Miyai et al., 2024), proposing that internal, pixel-level patterns of degradation offer a more direct and effective regularizer for enhancing robustness without relying on external semantic supervision (Zhou et al., 2022).

### 2.2  FROM POST-HOC EXPLANATION TO EXPLANATION-AS-SUPERVISION

Traditionally, eXplainable AI (XAI) methods like Grad-CAM (Selvaraju et al., 2017) have been used post-hoc to interpret model decisions, but recent work in Self-eXplainable AI (S-XAI) or explanation-guided training (Hou et al., 2024; Qing et al., 2022) has shifted towards using explanations as direct supervision. Prior studies have shown that training with human-provided saliency

maps improves generalization (Ismail et al., 2021), interpretability in medical imaging, and robustness against adversarial attacks (Guesmi et al., 2024; Mehra, 2020; Boopathy et al., 2020; Noack et al., 2021). Building on this trend, our work introduces two contributions: first, we generate localized corruption signals in a fully self-supervised way from data corruption rather than relying on costly annotations; second, we frame explanation guidance as a structural regularizer specifically aimed at improving out-of-distribution robustness in large-scale VLMs. By training the model to produce fine-grained rationales for distinguishing clean versus corrupted data, we encourage it to develop deeper, more resilient representations without external supervision.

## 3 METHODOLOGY

Our proposed method, Corruption-Guided Finetuning, trains a Vision-Language Model (VLM) to build a more robust and detailed model of data integrity specifically data corruption. The core of our approach is to solve the technical challenge of integrating a global classification objective with a local, dense prediction objective. To achieve this stability, we introduced a model architecture capable of both tasks, combined with a three-stage training paradigm that prevents the two learning signals from interfering.

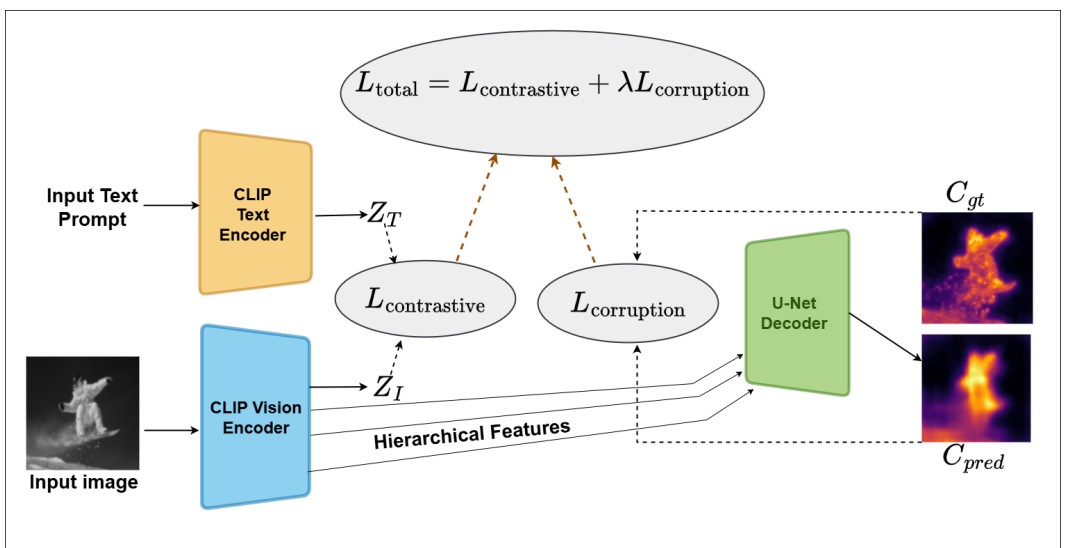

Figure 1: The CG-CLIP Architecture. Our model augments a standard CLIP Vision Transformer (ViT) backbone with a U-Net-style decoder for dense corruption mapping. The CLIP Text Encoder processes Input Text Prompts to generate text embeddings ($Z_T$). The CLIP ViT Backbone processes the Input Image ($I_x$) to produce a global image embedding ($Z_I$) from its final layer for contrastive learning. Simultaneously, it provides multi-scale features to the decoder. The U-Net Decoder takes these hierarchical features via skip connections to predict a dense corruption map ($C_{pred}$), which is compared against the ground truth corruption map ($C_{gt}$) using $L_{corruption}$. Both $L_{contrastive}$(between $Z_I$ and $Z_T$) and the weighted $L_{corruption}$ contribute to the total loss ($L_{total}$), guiding the model to build a robust internal model of natural image statistics.

### 3.1 CG-CLIP ARCHITECTURE

The core of our model, CG-CLIP, whose architecture is illustrated in Figure 1, is a pre-trained CLIP (ViT-B/32) (Radford et al., 2021) that we architecturally modify to perform both global classification and local pixel-level corruption prediction. This is achieved by augmenting the ViT backbone with a lightweight decoder designed for dense prediction tasks. This architectural modification is precisely designed to move beyond CLIP's global, object-centric features to capture fine-grained structural information. We attach a U-Net-style decoder (Ronneberger et al., 2015) to the ViT backbone, a choice motivated by its proven effectiveness in dense prediction tasks like semantic segmentation (Krithika Alias AnbuDevi & Suganthi, 2022; Huang et al., 2020; Wang et al., 2023b). The U-Net's

signature architecture, which uses skip connections to progressively combine deep, semantic feature maps with shallow, high-resolution ones, is ideally suited for our purpose. It allows the model to synthesize the high-level contextual understanding of what constitutes a corruption (from deep layers) with the precise spatial information needed to localize it (from shallow layers).

A standard ViT, however, processes an image into a single [CLS] token for global representation, which discards the spatial information necessary for a dense task. To provide the decoder with the required multi-scale inputs, we adopt a hierarchical feature extraction strategy, a principle validated in works that adapt transformers for dense prediction (Wang et al., 2021; Rao et al., 2022). We intercept the sequence of patch tokens from the ViT's transformer blocks at multiple depths: The final block provides high-level semantic context, crucial for understanding the image content. The intermediate block captures more complex, part-based information. A shallow block retains the fine-grained spatial details essential for sharp, accurate localization(Lin et al., 2017). Please refer to Appendix A.1 for more details. These multi-scale features are then fed into the U-Net decoder via skip connections at each upsampling stage. This architecture effectively repurposes the ViT from a simple classifier into a powerful, multi-scale feature extractor, enabling the model to produce high-fidelity corruption maps without compromising its global feature learning.

## 3.2 Ground Truth Corruption Map Generation

A key component of our method is a reliable ground truth signal for the corruption prediction task. We generate this target by computing the perceptual difference between a clean image $I_c$ and its corrupted version $I_x$. To generate a stable and perceptually meaningful ground truth signal, we use the pre-trained VGG-16 network (Simonyan & Zisserman, 2014). This choice aligns with established perceptual metrics like LPIPS (Zhang et al., 2018) and DISTS (Ding et al., 2020), which similarly leverage the features of a pre-trained VGG network to effectively model human judgments of image similarity. Its multi-level features allow us to capture a rich spectrum of differences, from low-level textures to higher-level structural changes. We build upon this established technique for measuring perceptual similarity, allowing us to focus our contribution on our proposed training framework that leverages these maps for OOD regularization. For a comprehensive visual comparison of various corruption map ground truth generation methods, and a detailed justification for our chosen approach, please refer to Appendix A.3. Let $V_l(I)$ be the feature map from the $l^{\text{th}}$ layer of the VGG network for an input image $I$. We extract features from a set of layers $L$, where $l \in L$ and $n = |L|$. To ensure spatial consistency for averaging, we define an upsampling operator $\mathcal{U}_l$ that resizes the feature map from layer $l$ to the spatial dimensions of the input image. The ground truth corruption map, $C_{\text{gt}}$, is then generated by computing the average pixel-wise $L_1$ distance across the upsampled feature maps from these layers. The value at each pixel location $(i, j)$ is formally defined as:

$$C_{\text{gt}}(i, j) = \frac{1}{n} \sum_{l \in L} \left[ \mathcal{U}_l \Big( \|V_l(I_c) - V_l(I_x)\|_1 \Big) \right] (i, j) \qquad (1)$$

We use the $L1$ distance as it is known to be less sensitive to large outlier differences and often encourages sharper, less blurry predictions compared to the $L2$ norm, making it well-suited for saliency tasks. This multi-level feature comparison provides a robust target that captures both low-level textural differences and higher-level structural changes.

## 3.3 The "Adapt-Isolate-Tune" Training Pipeline

A central challenge in our approach is to effectively integrate two distinct learning objectives: a global, spatially-invariant classification task and a local, pixel-wise corruption task. Rather than relying on simultaneous optimization with complex gradient management techniques (Sener & Koltun, 2018; Chen et al., 2018; Yu et al., 2020; Qin et al., 2025), we propose a structured curriculum learning strategy, which we call the "Adapt-Isolate-Tune" pipeline. This paradigm is built on the hypothesis that knowledge should be acquired sequentially. The model first learns a robust global representation for the primary task before that representation is refined using the rich, structured feedback from the dense auxiliary task. This staged approach ensures that the powerful regularizing effect of the pixel-level prediction task enhances, rather than disrupts, the global feature learning.

**Stage 1: Adapt**

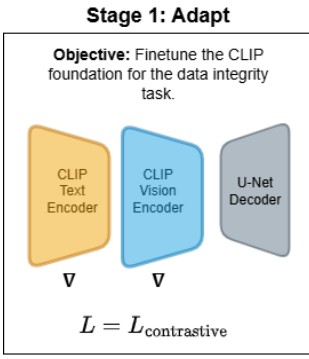

**Stage 2: Isolate**

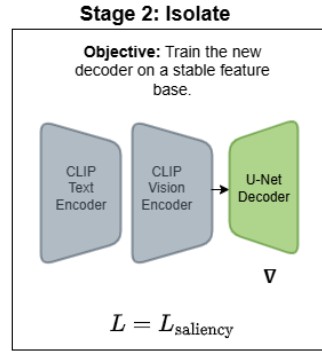

**Stage 3: Tune**

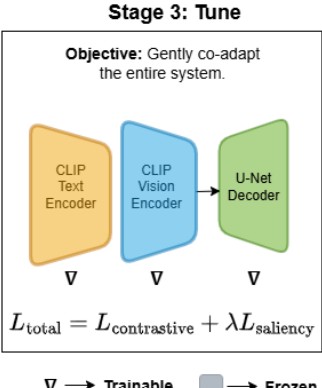

∇ → Trainable   ▢ → Frozen

Figure 2: The "Adapt-Isolate-Tune" Curriculum Learning Pipeline. Our principled three-stage strategy for training CG-CLIP. (Stage 1: Adapt) The model learns a foundational, global representation for data corruption. Both CLIP encoders are finetuned with a contrastive loss to distinguish clean vs. corrupt images. (Stage 2: Isolate) The model learns a specialized, local skill. With the encoders frozen to provide a stable feature base, only the U-Net decoder is trained to predict pixel-wise corruption maps. (Stage 3: Tune) The full system is trained end-to-end for co-adaptation. The now-competent decoder provides rich, structured feedback to the encoders, refining their global features with fine-grained spatial detail.

This staged approach provides a more stable and interpretable training process by first allowing the model to learn a robust global feature space before introducing and co-adapting it with the dense local task. This ensures that the powerful regularizing effect of the pixel-level corruption prediction task refines, rather than disrupts, the global feature learning. This staged approach is specifically designed to shift the model from its initial object-centric state towards a deep understanding of data corruption. To resolve the conflict between the global contrastive loss and the local corruption loss, we divide the training into three distinct stages, as illustrated in Figure 2:

**Stage 1: Adapt** In this foundational stage, the goal is to adapt the entire CLIP model's feature space to be sensitive to the coarse, global concept of data corruption. This stage begins the shift away from purely object-centric representations by exposing it to corruption in a low-information way initially, but sets the stage for richer learning. We finetune both the image and text encoders using the standard contrastive loss, following the FLYP methodology. Given a batch of $N$ clean images $I_c$ and $N$ corrupted images $I_x$, we construct a concatenated batch of size $B = 2N$, $I = [I_c, I_x]$, with corresponding text prompts $T = [T_{clean}, T_{corrupt}]$. The prompts we use, such as "a clean photograph" and "a corrupt photograph", are intentionally object-agnostic. This strategy, also leveraged in works like AnomalyCLIP (Zhou et al., 2023) for zero-shot anomaly detection, encourages the model to learn a general concept of data corruption that is independent of any specific object's features. Let $Z_I$ and $Z_T$ be the normalized global image and text embeddings for a batch of $B$ corresponding pairs, respectively. The model is trained to maximize the cosine similarity of true pairs. The similarity score between the $i^{th}$ image embedding and the $j^{th}$ text embedding is denoted as $s_{ij}$. The loss function for this stage is the symmetric cross-entropy over the similarity logits, scaled by a learnable temperature parameter $\tau$, as shown in Equation (2):

$$L_{\text{contrastive}} = -\frac{1}{2B} \left( \sum_{i=1}^{B} \log \frac{\exp(s_{ii}/\tau)}{\sum_{j=1}^{B} \exp(s_{ij}/\tau)} + \sum_{j=1}^{B} \log \frac{\exp(s_{jj}/\tau)}{\sum_{i=1}^{B} \exp(s_{ij}/\tau)} \right), \quad (2)$$

**Stage 2: Isolate**

The goal of this stage is to calibrate the decoder's gradient signal before it is used to regularize the shared encoder. To achieve this, the adapted CLIP encoder from Stage 1 is frozen, providing a stable feature base and avoiding conflicting gradients. This isolation ensures the new decoder becomes competent at its dense prediction task. Consequently, when joint training begins in Stage 3, the decoder provides a structured, meaningful gradient signal that refines the encoder, rather than the

noisy, disruptive gradients from a randomly initialized decoder. We use the Binary Cross-Entropy (BCE). The optimizer updates only the decoder's parameters to minimize the pixel-wise BCE loss between its predicted corruption map and the ground truth. The model is trained to minimize the pixel-wise difference between its predicted corruption map $C_{pred}$ and the VGG-based ground truth map $C_{gt}$. Let $M$ be the total number of pixels in the corruption map, and let $c_{pred,i}$ and $c_{gt,i}$ be the predicted logit and ground truth value for the $i^{th}$ pixel, respectively. The corruption loss is defined in Equation (3):

$$L_{\text{corruption}} = -\frac{1}{M} \sum_{i=1}^{M} [c_{\text{gt},i} \cdot \log(\sigma(c_{\text{pred},i})) + (1 - c_{\text{gt},i}) \cdot \log(1 - \sigma(c_{\text{pred},i}))] \tag{3}$$

**Stage 3: Tune** In the final stage, the entire system is unfrozen for end-to-end finetuning, allowing for co-adaptation. The now competent decoder provides rich, structured gradient feedback to the encoder. This allows the harder, dense prediction task to refine the global feature representation, compelling it to preserve the fine-grained structural details necessary for robust corruption detection and leading to the significant observed performance gains. The full model (CLIP visual and textual encoder and U-Net decoder) is unfrozen. The combined loss and differential learning rates carefully balance preserving CLIP's existing semantic power while integrating the new structural understanding. The total loss for Stage 3 combines both the contrastive and corruption mapping objectives, allowing for a balanced co-adaptation of the model components, as expressed in Equation (4):

$$L_{\text{total}} = L_{\text{contrastive}} + \lambda L_{\text{corruption}} \tag{4}$$

where $\lambda$ is a hyperparameter to balance the two tasks. To prevent catastrophic forgetting of the robust global features learned in Stage 1, we employ differential learning rates. The CLIP backbone is updated with a significantly lower learning rate to preserve its powerful pre-trained features, while the newer U-Net decoder is trained with a higher learning rate to encourage faster convergence. This ensures that the powerful pre-trained features are preserved while being subtly refined by the new, fine-grained corruption-mapping objective. The implementation details and hyperparameters for each stage are described in Appendix A.1.

## 4 EXPERIMENTS AND RESULTS

To validate our hypothesis that corruption-guided finetuning enhances corruption robustness and generalization across domain shifts, we conduct a series of quantitative and qualitative experiments. We aim to answer three key questions: (1) How does the feature space of the CLIP vision encoder evolve throughout our three-stage training paradigm? (2) How does the final CG-CLIP model compare to strong finetuning baselines on both in-domain and OOD datasets? (3) Does the model learn a generalizable concept of corruption, applicable even to unseen artifact types?

### 4.1 DATASETS AND CORRUPTIONS

For training, we use the Microsoft COCO 2017 dataset (Lin et al., 2014), chosen for its large scale ($118, 287$ images) and diverse, high-quality images that define a "clean" baseline. Each image is dynamically corrupted with noise, blur, weather, or digital artifacts from (Hendrycks & Dietterich, 2019), with severity randomly sampled from 1 to 5, exposing the model to a wide spectrum of degradations. To evaluate robustness, we employ two datasets. CUB-200-2011 (Wah et al., 2011), with $11, 800$ high-quality bird photographs, serves as a clean-data benchmark, confirming that gains in out-of-distribution (OOD) robustness do not compromise accuracy on uncorrupted images. Caltech-256 (Griffin et al., 2007), containing $30, 600$ images across 257 categories, including many older, scanned, and lower-resolution images, functions as a critical OOD stress test. Strong performance here indicates the model has learned generalizable concept of data corruption, overcoming biases inherent in modern datasets like COCO. Please refer to the Appendix A.1 for the list of seen and unseen corruptions.

## 4.2 T-SNE ANALYSIS: VISUALIZING FEATURE SPACE TRANSFORMATION

We first analyze the evolution of the CLIP vision encoder's feature space by visualizing the embeddings of clean and corrupted images using t-SNE (Maaten & Hinton, 2008). Figure 1 presents a comparison of the feature space at three key milestones.

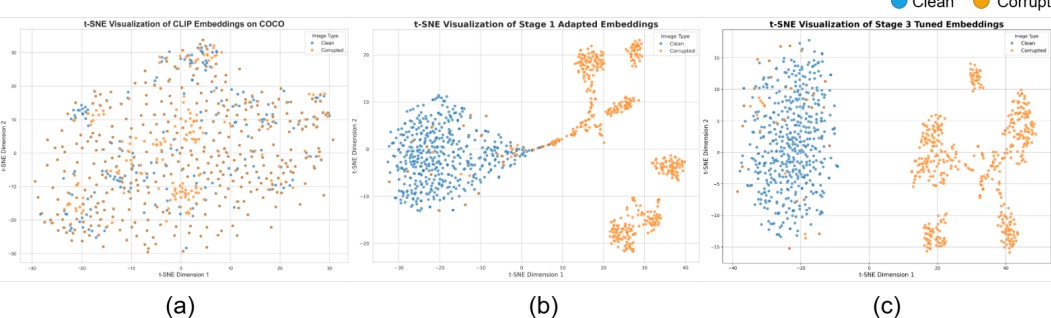

(a)                                    (b)                                    (c)

Figure 3: t-SNE visualization of the feature space. (a) The frozen, off-the-shelf CLIP encoder shows no meaningful separation between clean (blue) and corrupted (orange) images. (b) After Stage 1 (Adapt), the feature space is well-structured, separating the classes into distinct clusters. (c) After Stage 3 (Tune), the separation becomes even more pronounced, with tighter clusters and a wider margin.

(i) Original CLIP: The frozen CLIP encoder's failure to separate clean from corrupted images (Figure 3, (a)) provides direct evidence of its inherent object-centric bias. Because the model is trained to recognize objects, it learns to discard the very structural patterns needed to assess data corruption. Consequently, its embeddings are low-information for this task, causing the feature clusters to completely overlap.

(ii) Stage 1 Model (FLYP): After the "Adapt" stage (Figure 3, (b)), the feature space undergoes a significant transformation. The FLYP contrastive objective successfully organizes the embeddings, pulling clean and corrupted images apart into two distinct and largely separable clusters. This demonstrates that the model has learned a robust, global representation for the concept of data corruption, forming a strong baseline.

(iii) Stage 3 Model (CG-CLIP): After the final "Tune" stage, the model learns a highly discriminative feature space that robustly separates the two classes (Figure 3, (c)). The clusters become significantly more compact and the margin between them widens, providing a stark contrast to the original model's failure. This result validates our central hypothesis: the dense structural guidance provided by the corruption mapping objective is the direct cause of this improvement. By forcing the encoder to provide fine-grained features for the pixel-wise corruption mapping task, the gradients from the decoder act as a powerful structural regularizer. This compels the model to move beyond its initial object-centric bias and learn the very features of data corruptions it was pre-trained to ignore, resulting in a more robust and fundamentally superior representation for this task.

## 4.3 QUALITATIVE ANALYSIS: DECODER PERFORMANCE AND ITS IMPACT

The refinement observed in the feature space is a direct consequence of the decoder's performance during joint training. As shown in Figure 4, the U-Net decoder, trained during Stage 2 and refined in Stage 3, learns to produce corruption maps that are a high-fidelity match to the ground truth.

To generate such accurate maps, the decoder requires the encoder to provide it with features that retain precise spatial information. During the "Tune" stage, the gradients from this local corruption loss flow back to the encoder. This pressure to preserve the fine-grained, structural "fingerprints" of corruption prevents the encoder from discarding too much spatial information in favor of a purely abstract global representation. It is this mechanism, the need to serve a competent decoder, that drives the final refinement of the feature space, enhancing its structure and leading to superior OOD classification.

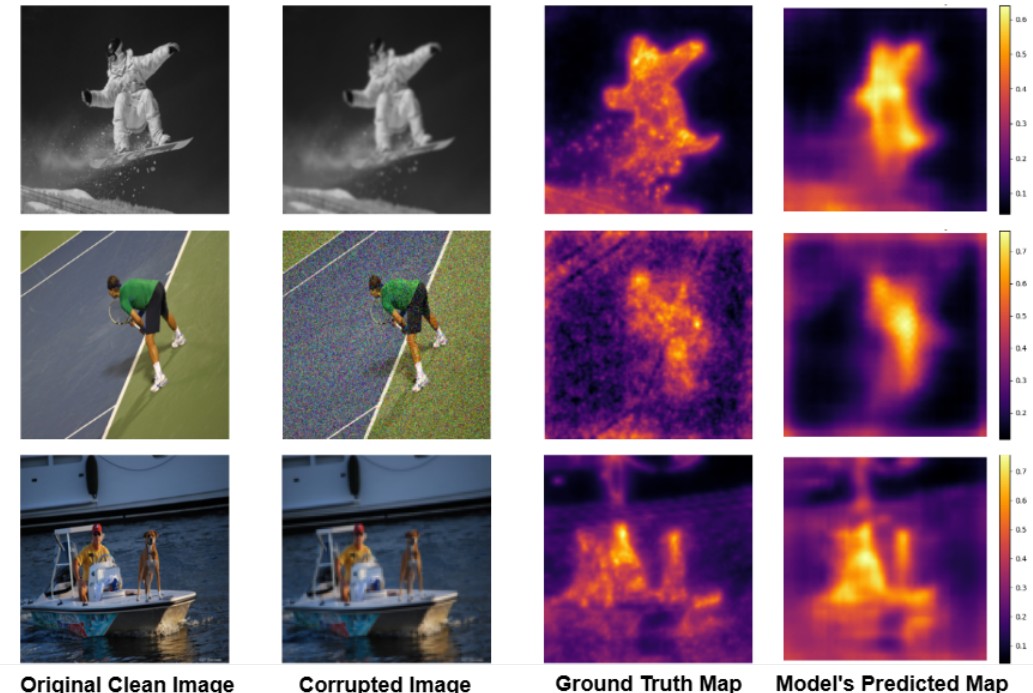

**Original Clean Image**   **Corrupted Image**   **Ground Truth Map**   **Model's Predicted Map**

Figure 4: Example outputs of the CG-CLIP decoder after Stage 3 for different corruptions (Gaussian blur; severity 4, Gaussian noise; severity 3, Pixelate; severity 2) . The predicted corruption maps closely match the ground truth, effectively localizing corruptions. This demonstrates the decoder's competence, the mechanism that drives the encoder's feature space refinement during the "Tune" stage.

### 4.4 QUANTITATIVE RESULTS

Our quantitative analysis, presented in Table 1, validates the effectiveness of our proposed CG-CLIP model and the "Adapt-Isolate-Tune" training pipeline across both in-domain (ID) and out-of-distribution (OOD) datasets. We first establish the baselines. A simple CLIP + Linear Probe struggles significantly on the OOD Caltech-256 stress test, achieving only 65.00% accuracy, which highlights the object-centric bias of the original model. Our Stage 1 Baseline (FLYP) adapts the model for the data integrity task, substantially improving the OOD accuracy to 88.64%. Our ablation studies confirm the necessity of our three-stage curriculum. An end-to-end joint training approach, which omits the staged curriculum, is hampered by conflicting gradients, reaching an accuracy of 85.79% on Caltech-256. Crucially, removing the final "Tune" stage (Ours (w/o Tune stage)) yields results identical to the FLYP baseline (88.64% accuracy), demonstrating that the joint co-adaptation in Stage 3 is essential for transferring the learned structural knowledge from the decoder back to the encoder. Our full model, CG-CLIP, significantly outperforms all baselines and ablations. On the challenging OOD Caltech-256 dataset, CG-CLIP achieves an accuracy of 97.45%, representing a substantial 8.81 percentage point improvement over the strong FLYP baseline. This performance leap is consistent across all metrics, with AUROC and F1-Score also improving by over 8 points to 97.50% and 97.49%, respectively. Furthermore, CG-CLIP also achieves near-perfect, state-of-the-art performance on the ID datasets, reaching 99.77% accuracy on COCO and 98.68% on CUB-200. These results strongly validate our hypothesis that integrating a dense, structural task via the "Adapt-Isolate-Tune" pipeline effectively mitigates object-centric bias and produces a more robust, generalizable feature representation. Importantly, this significant performance gain is achieved with no additional computational cost at inference time, as the auxiliary decoder used during training is discarded for the final classification task.

Table 1: Accuracy in detecting corrupted images on in-domain (ID) and out-of-distribution (OOD) datasets. Our CG-CLIP model significantly outperforms both baselines, especially on the challenging Caltech-256 stress test. Note that our 'Stage 1 Baseline', i.e., Adapt stage, adheres to the implementation approach described in the state-of-the-art FLYP method (Goyal et al., 2023).

| Method | COCO | | | CUB-200 | | | Caltech-256 | | |
|---|---|---|---|---|---|---|---|---|---|
| | Accuracy | AUROC | F1-Score | Accuracy | AUROC | F1-Score | Accuracy | AUROC | F1-Score |
| CLIP + Linear Probe | 95.14% | 95.21% | 95.16% | 95.03% | 95.16% | 95.12% | 65.00% | 65.09% | 65.07% |
| FLYP (Adapt Stage) | 95.26% | 95.40% | 95.18% | 95.18% | 95.33% | 95.25% | 88.64% | 88.85% | 89.02% |
| End-to-End Joint Training | 94.60% | 94.72% | 94.19% | 92.93% | 92.42% | 92.51% | 85.79% | 86.67% | 85.22% |
| Ours (w/o Isolate stage) | 97.57% | 97.74% | 97.59% | 96.87% | 97.32% | 96.25% | 92.79% | 93.18% | 92.59% |
| Ours (w/o Tune stage) | 95.26% | 95.40% | 95.18% | 95.18% | 95.33% | 95.25% | 88.64% | 88.85% | 89.02% |
| CG-CLIP (Ours) | **99.77%** | **99.79%** | **99.77%** | **98.68%** | **98.68%** | **98.70%** | **97.45%** | **97.50%** | **97.49%** |

## 4.5 GENERALIZATION TO UNSEEN CORRUPTIONS

To further test the generalization capabilities of CG-CLIP, we evaluated its performance on a held-out set of corruption types that the model had never encountered during training. The results in Table 2 are a testament to the model's robust learning. CG-CLIP maintains an accuracy of over 91% across all datasets. This provides strong evidence against overfitting to specific artifact patterns. By being forced to learn the local "fingerprints" of corruption through the dense prediction task, the model has developed a deeper, more fundamental understanding of what constitutes a deviation from a "clean" image, allowing it to successfully identify corruptions it has never seen before. This is a key indicator that the model has learned more than a simple classification boundary; it has moved beyond simple classification to develop a robust model of data corruption, allowing it to detect artifacts far more reliably.

Table 2: Performance on a held-out set of unseen corruption types. CG-CLIP maintains high accuracy across all datasets, indicating it has learned a generalizable model of corruption.

| Method | COCO | | | CUB-200 | | | Caltech-256 | | |
|---|---|---|---|---|---|---|---|---|---|
| | Accuracy | AUROC | F1-Score | Accuracy | AUROC | F1-Score | Accuracy | AUROC | F1-Score |
| FLYP (Adapt Stage) | 90.40% | 91.70% | 89.80% | 87.34% | 90.87% | 86.34% | 84.54% | 89.29% | 82.92% |
| CG-CLIP (Ours) | **92.3%** | **92.6%** | **91.7%** | **91.5%** | **91.6%** | **90.9%** | **91.5%** | **91.7%** | **91.1%** |

## 5 CONCLUSION

In this work, we addressed the inherent object-centric bias of large vision-language models by introducing Corruption-Guided Finetuning. We demonstrated that a dense, structural auxiliary task, when integrated via our principled "Adapt-Isolate-Tune" curriculum learning strategy, can effectively regularize the model to learn robust, structural features. This methodology led our model, CG-CLIP, to achieve a remarkable $\sim 9$ percentage point accuracy gain on the challenging Caltech-256 out-of-distribution benchmark. This is achieved with a temporary 5.96% increase in trainable parameters during training; because the auxiliary decoder is discarded, the deployed model has no additional inference overhead. While our experiments focused on common corruptions, this work validates a powerful approach for enhancing VLM robustness. Future work should extend this paradigm to a wider spectrum of data integrity challenges, including adversarial attacks and GAN-generated artifacts. Finally, the continuous-valued corruption score enables novel replay buffer management in Continual Learning and suggests future exploration in safety-critical domains such as medical imaging and autonomous navigation.

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

# A    APPENDIX

## A.1    IMPLEMENTATION DETAILS

All experiments were conducted using PyTorch on a single RTX NVIDIA 6000 ADA GPU . We utilize Automatic Mixed Precision (AMP) with torch.cuda.amp. GradScaler for improved computational efficiency and reduced memory footprint across all training stages.

Model and Preprocessing- Our CG-CLIP model is built upon the pre-trained ViT-B/32 CLIP model provided by OpenAI. For the U-Net decoder, we extract multi-scale features from the 4th, 8th, and 12th transformer blocks of the ViT backbone. The decoder itself consists of three upsampling stages with feature channel dimensions of [384, 192, 96]. The ground truth corruption maps are generated using a frozen, pre-trained VGG-16 network with weights from ImageNet. We extract features from a set of layers $L \subseteq \{1, 2, 3, 4, 5\}$. All input images are resized to 224x224 pixels using bicubic interpolation before being processed by their respective models.

Training Configuration. We employ the AdamW optimizer (Loshchilov & Hutter, 2017) for all training stages. We use a batch size of 128. Gradient clipping with a max norm of 1.0 is used for added stability.

During training, we dynamically apply corruptions to each image, adapted from the corruption benchmark (Hendrycks & Dietterich, 2019). The set of seen corruptions the model was trained on includes noise (Gaussian, speckle), blur (motion, glass, Gaussian), weather (fog, spatter), and digital corruptions (pixelate, brightness). To test for generalization, we evaluated the model on a held-out set of unseen corruptions that were never encountered during any training stage. This unseen set included noise (impulse, shot), blur (defocus, zoom), weather (snow, frost), and a digital corruption (saturate).

Our three-stage training pipeline is configured as follows:

- Stage 1 (Adapt): The CLIP model is finetuned for 10 epochs using the contrastive loss. We use the AdamW optimizer with a learning rate of $1e{-}6$, and other hyperparameters ($\beta_1 = 0.9, \beta_2 = 0.98, \epsilon = 1e{-}6, weight\_decay = 0.2$) as recommended in the original CLIP paper.
- Stage 2 (Isolate): The U-Net decoder is trained for 5 epochs with the CLIP backbone frozen. We use a dedicated decoder learning rate of $2e{-}4$
- Stage 3 (Tune): The full CG-CLIP model is trained for 5 epochs. We use differential learning rates: the CLIP backbone is updated with a learning rate of $1e{-}6$ , while the decoder continues to use the learning rate of stage 2. The corruption loss weight ($\lambda$) for the combined loss was set to 1.5, a value determined empirically via a grid search on a held-out validation split of the training data.

## A.2    EVALUATION METRICS

To comprehensively evaluate the performance of our models on the binary classification task of corruption detection, we selected three standard and complementary metrics: Accuracy, F1 Score, and the Area Under the Receiver Operating Characteristic Curve (AUROC). Each metric provides a unique perspective on the classifier's performance and, when taken together, allows for a more complete assessment of robustness across datasets. Empirical results for these metrics are reported in Table 1.

### A.2.1    ACCURACY

Accuracy is the most intuitive performance measure and is defined as the ratio of correctly classified instances (both clean and corrupted images) to the total number of instances in the dataset. While straightforward, it provides a valuable top-level assessment of the model's overall correctness. It is calculated as in Equation 5.

$$\text{Accuracy} = \frac{TP + TN}{TP + TN + FP + FN} \tag{5}$$

**True Positives (TP):** Corrupted images correctly identified as corrupted.

**True Negatives (TN):** Clean images correctly identified as clean.

**False Positives (FP):** Clean images incorrectly identified as corrupted.

**False Negatives (FN):** Corrupted images incorrectly identified as clean.

As shown in Table 1, Accuracy captures the strong in-domain performance of all models (above 94% on COCO and CUB-200). However, on the more challenging OOD setting of Caltech-256, baselines degrade sharply (e.g., CLIP + Linear Probe at only 65.00%), while our proposed CG-CLIP achieves a robust 97.45%, demonstrating superior generalization ability.

### A.2.2 F1 SCORE

The F1 Score is the harmonic mean of Precision and Recall, providing a more robust measure than Accuracy, especially when the class distribution might be uneven. It balances the trade-off between identifying all positive instances (Recall) and ensuring that the identified instances are truly positive (Precision). Precision and Recall are defined in Equation 6 and Equation 7, and the F1 Score is calculated using Equation 8.

$$\text{Precision} = \frac{TP}{TP + FP} \tag{6}$$

$$\text{Recall} = \frac{TP}{TP + FN} \tag{7}$$

$$\text{F1 Score} = 2 \times \frac{\text{Precision} \times \text{Recall}}{\text{Precision} + \text{Recall}} \tag{8}$$

From Table 1, the F1 Score closely follows Accuracy trends but provides an additional safeguard against misleading results in skewed scenarios. Notably, while end-to-end joint training achieves 85.22% F1 on Caltech-256, CG-CLIP surpasses it with 97.49%, highlighting its ability to minimize both false positives and false negatives.

### A.2.3 AREA UNDER THE ROC CURVE (AUROC)

The AUROC score measures the ability of a classifier to distinguish between classes. It is a threshold-independent metric that summarizes performance across all possible classification thresholds. The ROC curve plots the True Positive Rate (TPR) against the False Positive Rate (FPR) at various thresholds, as defined in Equations 9 and 10.

$$\text{TPR} = \frac{TP}{TP + FN} \tag{9}$$

$$\text{FPR} = \frac{FP}{FP + TN} \tag{10}$$

An AUROC score of 1.0 represents a perfect classifier, while a score of 0.5 indicates no discriminative ability beyond random chance. As shown in Table 1, AUROC highlights the robustness gap most clearly: CLIP + Linear Probe achieves only 65.09% on Caltech-256, whereas CG-CLIP reaches 97.50%. This confirms that our method consistently separates clean and corrupted images across thresholds, avoiding overfitting to a specific decision boundary.

### A.3 COMPARATIVE ANALYSIS OF CORRUPTION MAP GROUND TRUTH METHODS

In defining our auxiliary corruption prediction task, a critical decision involved selecting an appropriate metric to generate ground truth maps that accurately reflect structural changes due to corruption. We evaluated several common image difference metrics: pixel-wise L1, LPIPS (Learned Perceptual Image Patch Similarity) (Zhang et al., 2018), DISTS (Deep Image Structure and Texture

Similarity) (Ding et al., 2020), and L1 applied to features extracted from a pre-trained VGG network. Visual comparisons, exemplified in Figure 5, Figure 6, and Figure 7, clearly highlight the rationale for our choice of L1 on VGG features.

**Pixel-wise L1 (Panel 3):** As seen in Panel 3 of Figure 5, Figure 6, and Figure 7, a direct pixel-wise L1 difference between the clean and corrupted image is extremely noisy and highly sensitive to minor, high-frequency variations. While it technically highlights changed areas, the resulting map is chaotic and provides very little coherent structural information. This low-level representation would provide a poor, unstable signal for a learning objective.

**LPIPS (Panel 4):** LPIPS, designed to mimic human perception, produces smoother, perceptually-aware difference maps (Panel 4 of Figure 5, Figure 6, and Figure 7). It correctly emphasizes regions where human perception would notice a change. However, for our specific goal of localizing data integrity corruptions, LPIPS can sometimes be too smooth or too focused on "perceptual quality". In cases like "motion_blur" (Panel 4 of Figure 5, Figure 6), while it broadly indicates motion, it might overly smooth the boundaries or fail to precisely localize the most structurally impacted regions at a fine-grained level important for feature learning. For "pixelate" (Panel 4 of Figure 7), it clearly identifies the affected area but may not offer the sharpest delineation of where the core structural information resides.

**DISTS (Panel 5):** DISTS (Panel 5 of Figure 5, Figure 6, and Figure 7) similarly aims for perceptual relevance by considering structural and textural differences. Its maps are generally smooth and highlight corrupted regions well, often providing a slightly different emphasis than LPIPS. However, similar to LPIPS, it can sometimes produce maps that are overly diffused or less focused on the precise object boundaries and internal structures impacted by the corruption, especially at the feature level. For example, in the "motion_blur" examples, DISTS provides a broad area of change but might lack the distinctness needed for structural identification.

**L1 on VGG Features (Panel 6):** This metric consistently produces corruption maps that are **structurally coherent, less noisy than pixel-L1, and more focused on meaningful object boundaries and internal components than LPIPS or DISTS** (Panel 6 of Figure 5, Figure 6, and Figure 7).

- For "motion_blur" (Panel 6 of Figure 5, Figure 6), it effectively highlights the moving subjects (the tennis player, the giraffes) and their immediate surroundings where motion blur is most pronounced, while largely suppressing less relevant background changes.
- For "pixelate" (Panel 6 of Figure 7), it sharply localizes the affected figure, delineating its form with greater precision than the other perceptual metrics.

Applying L1 difference to features from a pre-trained VGG network captures perceptual differences at various levels of abstraction, from edges and textures to more complex shapes. The VGG, being a strong image feature extractor, already provides a hierarchical representation. Taking L1 between these feature maps (rather than raw pixels) allows us to quantify differences in structural content in a robust, perceptually aligned manner, while also being sufficiently precise to guide a dense prediction task. This approach effectively filters out irrelevant high-frequency noise (unlike pixel-L1) and focuses on the structural disruptions caused by corruption, without being overly abstract or diffuse (as LPIPS/DISTS can sometimes be for this specific task). This provides the most stable, yet information-rich, signal for our decoder to learn from.

Therefore, L1 on VGG features was chosen as the most appropriate method for generating our ground truth corruption maps, providing robust and structurally-aware targets that are crucial for effectively regularizing our Corruption-Guided Finetuning paradigm.

### A.4 LLM USAGE

We used large language models (LLMs) in a limited capacity to support this work. Specifically:

1. **Writing assistance:** LLMs were employed to improve the clarity, grammar, and readability of the manuscript. No content generation, experimental design, or technical claims were delegated to LLMs.

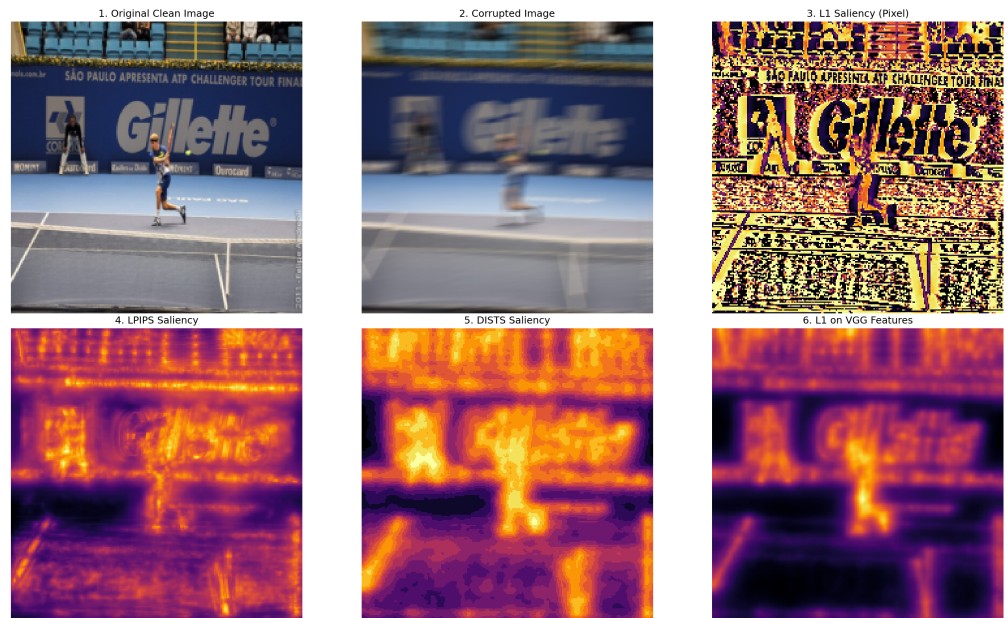

Figure 5: Ground Truth Comparison for "motion_blur" (Example 1). This figure shows various methods for generating corruption maps from the difference between an original clean image (Panel 1) and a corrupted image (Panel 2). Panel 3: Pixel-wise L1 Map. Panel 4: LPIPS Map. Panel 5: DISTS Map. Panel 6: L1 on VGG Features Map (Our Ground Truth).

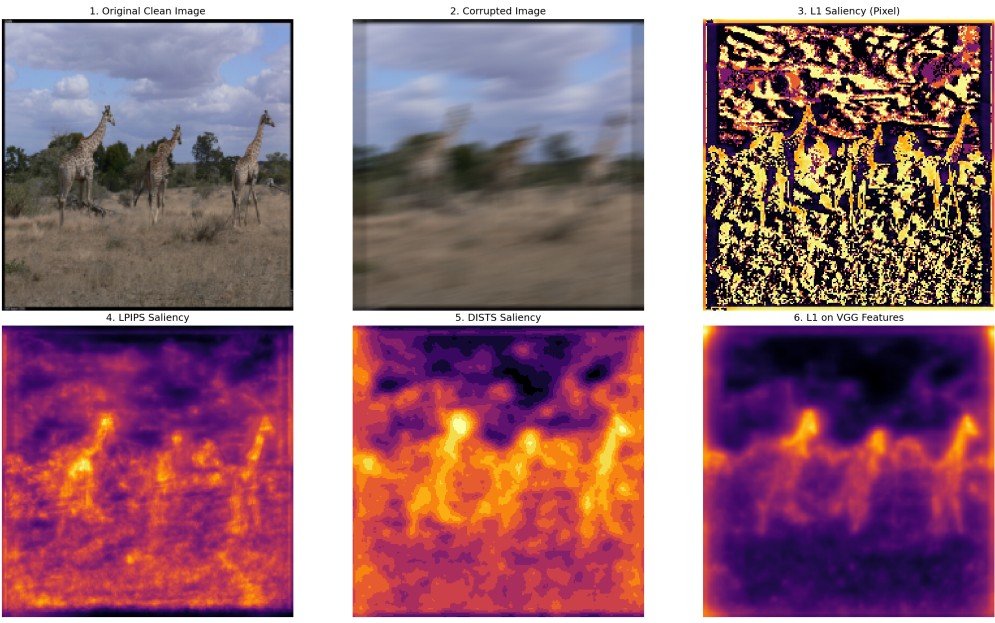

Figure 6: Ground Truth Comparison for "motion_blur" (Example 2). This figure shows various methods for generating corruption maps from the difference between an original clean image (Panel 1) and a corrupted image (Panel 2). Panel 3: Pixel-wise L1 Map. Panel 4: LPIPS Map. Panel 5: DISTS Map. Panel 6: L1 on VGG Features Map (Our Ground Truth).

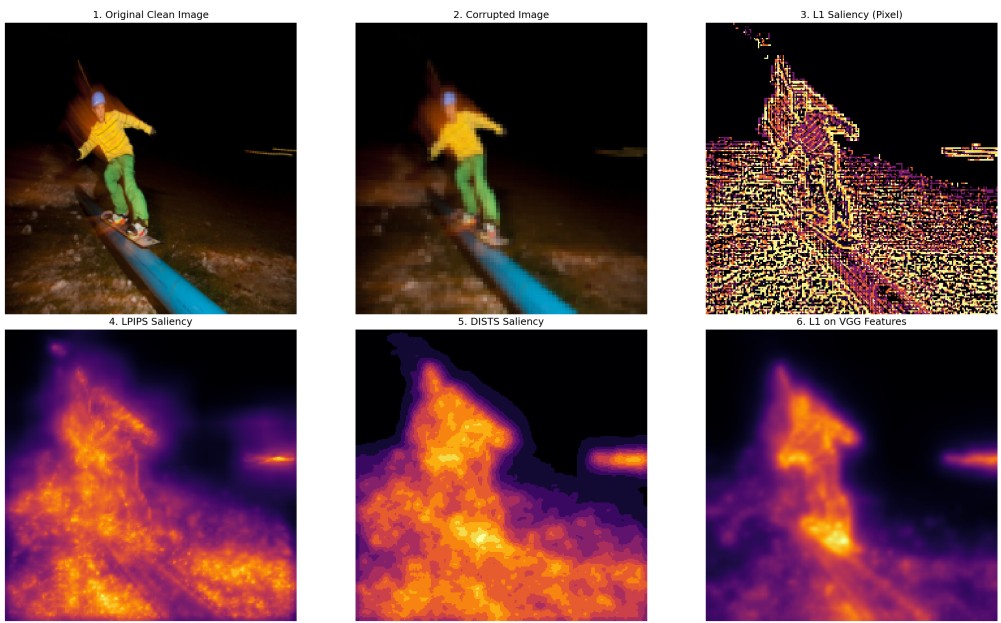

Figure 7: Ground Truth Comparison for "pixelate". This figure shows various methods for generating ground truth maps from the difference between an original clean image (Panel 1) and a corrupted image (Panel 2). Panel 3: Pixel-wise L1 Map. Panel 4: LPIPS Map. Panel 5: DISTS Map. Panel 6: L1 on VGG Features Map (Our Ground Truth).

2. **Literature search support:** LLMs were used as an auxiliary tool to help identify related works in the area of eXplainable AI (XAI). All references included in the paper were manually verified for accuracy and relevance by the authors.