# OpenReview forum: "CG-CLIP: Seeing Beyond Objects to Learn Corruption Robustness"
_ICLR.cc/2026/Conference — ICLR 2026 Conference Withdrawn Submission_

### Official Review · Reviewer_shkK · 2025-10-29

**Soundness:** 2
**Presentation:** 2
**Contribution:** 2
**Rating:** 2
**Confidence:** 4

**Summary:**

This paper proposes a training strategy to fine-tune CLIP models. The idea is to augment fine-tuning with the dense auxiliary task of predicting the difference between a clean and a corrupted version of an image. This difference is computed by encoding both images with VGG, and averaging over its (scaled) feature maps. Results show that their models do indeed outperform all other investigated models on the task of recognizing whether an image was corrupted or not, including strong baselines.

**Strengths:**

- The abstract idea of introducing a dense auxiliary task to support the semantic focus of the CLIP objective makes sense.
- The results on the classification tasks are good, I believe the claim that the proposed approach outperforms strong baselines on this specific task.
- It is promising that the approach generalizes to held-out ImageNet-C corruptions.

**Weaknesses:**

- The paper fails to show that the proposed auxiliary loss generalizes to fine-tuning settings other than detecting the local, idiosyncratic ImageNet-C style corruptions. The results for this particular setting are good, and I acknowledge that there is transfer between different corruption types, but these are quite similar and would all very clearly benefit from this specific auxiliary task. Overall, this whole approach seems like a good engineering strategy to solve the very specific problem of building a corrupted-vs-clean classifier for ImageNet-C style corruptions, but not like a general remedy for the object-centric bias of CLIP models identified in the abstract.
- Ideally, I would have liked to see, for example, that adding this auxiliary loss leads to bigger gains on fine-tuning CLIP models for ImageNet-classification (where FLYP also provides an independently verifiable baseline).
- The paper lacks clarity, in the sense that the actual task which is learned (a binary classification of images into corrupted / uncorrupted) is not sufficiently clearly stated. One might mistakenly believe that the authors train models to be robust to corruptions, i.e. classify images correctly into semantic categories in the presence of noise, but this is not the case. This is a problem of presentation rather than of substance, but the writing should be improved.
- I find the reference to interpretability a bit misleading; there is no real relation to interpretability concerns.
- I think that equation 1 contains a mistake: The upscaling operation is currently applied to the output of a norm, i.e. a scalar. I suspect that what the authors mean to say is that they take the pixel-wise absolute values of the differences and then upscale the result, to then take an average. Could the authors clarify this?
- The paper is generally sparse on details. This is improved by the Appendix, but information like e.g. which corruptions (or at least how many) were held out for table 2 should really be in the main paper.

Overall, I don't think this paper constitutes a sufficiently big contribution. The idea is interesting, but remains underexplored by only focusing on the very specific task of corruption detection, for which it is quite plausible that the proposed auxiliary task would work well, but which is not interesting in itself.

**Questions:**

- In stage 1, do you just map images to either of the two strings "a clean image" or "a corrupted image", or do you prepend these strings to the original captions of images?
- Not a question as such, but also too minor of a point to list it as a "weakness": The first sentence of the abstract does not seem quite correct. "Large pre-trained models like CLIP exhibit an object-centric bias, rendering them brittle for tasks like assessing robustness to common image corruptions." This implies that one would use CLIP models to assess the robustness (of some model, presumably) to common corruptions. But this is not what is done in the paper.

---

### Official Review · Reviewer_ncH9 · 2025-10-31

**Soundness:** 3
**Presentation:** 4
**Contribution:** 3
**Rating:** 6
**Confidence:** 4

**Summary:**

Proposes CG-CLIP, a corruption-robust CLIP finetuning that adds a dense auxiliary task: predicting pixel-wise corruption maps, trained with a three-stage Adapt --> Isolate --> Tune curriculum. The decoder is used only during training, then discarded for zero inference overhead. Reports strong OOD gains, especially on Caltech-256.

**Strengths:**

- Novel training strategy: clear and principled curriculum separates global contrastive learning from dense supervision, then co-adapts to avoid gradient interference.

- Clean deployment: auxiliary decoder is removed at test time, so no runtime cost increase.

- Compelling OOD results: substantial improvements over FLYP on Caltech-256 and strong ID performance on COCO and CUB-200.

- Qualitative support: decoder maps align with ground truth and are used to refine the encoder during Tune.

**Weaknesses:**

- Dataset protocol clarity: COCO is treated as ID, CUB-200, and Caltech-256 as OOD. The text states training on COCO with synthetic corruptions and evaluation on CUB-200 and Caltech-256, but it would benefit from explicitly discussing the corruptions used: why were these particular corruptions chosen, and how does choosing a subset of them during training affect the final performance?

- Stage scheduling rationale: Appendix provides specific choices (e.g., Stage-2 and Stage-3 epochs, differential learning rates, $\lambda$ chosen by grid search), but there is limited sensitivity analysis or justification of robustness to these hyperparameters.

- Ground-truth design choice: VGG-based perceptual differences are well argued against LPIPS/DISTS, yet it would be stronger to compare modern backbones for generating $C_{gt}$ (e.g., DINOv3 ConvNeXt/ViT features) to rule out dependence on dated features.

**Questions:**

- Dataset protocol and corruption subset:
1. What was the rationale for selecting this subset, and how sensitive are the results to adding or removing specific corruption families?
2. Does performance change materially if the training set covers all ImageNet-C types versus the chosen subset?

- Stage scheduling and hyperparameters:
1. How were the epoch counts for Adapt, Isolate, and Tune decided, and what ranges were explored?
2. How sensitive are the main results to the stage durations, differential learning rates, and the λ weight?


- Ground-truth map backbone:
1. Why is VGG preferred for generating $C_{gt}$ given stronger modern features exist?
2. Do DINOv3, ConvNeXt, or ViT features used in the same pipeline change outcomes?

---

### Official Review · Reviewer_4m6U · 2025-10-31

**Soundness:** 2
**Presentation:** 2
**Contribution:** 2
**Rating:** 2
**Confidence:** 4

**Summary:**

The paper proposes CG‑CLIP, a three‑stage training pipeline that augments CLIP with a U-Net decoder and a dense auxiliary task for mapping common corruptions. The authors argue that CLIP is “object‑centric,” so adding a dense, local corruption detection task regularizes against superficial correlations and improves robustness. Reported gains include a jump on Caltech‑256 from ~88% to 97.45% and improvements on COCO‑based OOD evaluation.

**Strengths:**

Using a dense auxiliary objective (predicting a corruption map) to regularize global alignment is intuitive and, if validated, would be a nice plug‑in to CLIP fine‑tuning. The paper presents this as the core contribution.

The Adapt-Isolate-Tune process is conceptually intuitive and modularizes the pipeline (contrastive alignment -> decoder training -> joint FT). That separation is useful for ablating the effect of each stage, which the authors perform in Table 1.

Provided the evaluation is well-established and the baselines are thorough, the claimed ~9% accuracy improvement on Caltech‑256 over strong CLIP baselines would be considerable.

**Weaknesses:**

Task definition is vague. It is unclear whether the intended task is (i) robust recognition, (ii) corruption detection, (iii) classification of corruption types, and/or (iv) corruption localization (maps). Different tasks imply different metrics, baselines, and training supervision, but the paper shifts between them and uses evidence from one to support claims about another.

Central hypothesis is inconsistent. The intro first frames the culprit as CLIP’s “object‑centric bias” and the classification objective producing superficial correlations. Later, the “central hypothesis” becomes that a "dense auxiliary task" is the key regularizer. The paper does not cleanly distinguish between biases from classification objectives vs. architectural choices, nor does it validate which is causally responsible.

“Object‑centric bias” is not substantiated. The paper does not cite prior work demonstrating that CLIP’s objective leads to discarding local cues. If anything, CLIP exposes patch tokens and can admit dense alignment (e.g., DenseCLIP), weakening the argument that it cannot encode locality.

t‑SNE is over‑interpreted. Low 2D t-SNE separability of CLIP embeddings is claimed to be "direct proof" and “unequivocal evidence" of object‑centric bias and CLIP failure. However, t‑SNE distances and cluster separations are not reliable evidence of true separability. A proper way to check for low separability is to train a probe (linear and non‑linear) on the embeddings. While the authors later report results for "CLIP + Linear Probe", the setting is under‑specified.

Architecture vs. objective is inconsistent. The method is pitched as classification objective‑level regularization, yet it introduces a U‑Net, hierarchical features, and VGG-extracted feature maps. It’s unclear which component is doing the heavy lifting (decoder capacity? number of hierarchical features? the dense loss?) because the necessary ablations are missing.

Missing ablations.
(a) CLIP-based localization: Use ViT patch tokens (instead of just [CLS]) and attention maps as localization cues (no U‑Net). If that works, the U-Net may be an unnecessary component.
(b) CLIP embedding differences: Since CLIP exposes hierarchical tokens, predicting differences of CLIP features between clean/corrupt could be a less expensive alternative to the VGG feature maps.
(c) Probes: Linear vs. MLP probes on CLIP features to test whether the base representation already separates clean vs. corrupt. The paper briefly shows a linear probe but without setup details, and no non‑linear probe (which might separate the representations better; see [1]).
(d) Freeze text encoder vs. tune both encoders: Following FLYP, the paper tunes both the vision and text encoders. It is also mentioned that the setup is similar to AnomalyCLIP, which may be an error since AnomalyCLIP freezes both encoders and instead performs prompt tuning. In any case, given that standard practice for fine-tuning CLIP involves freezing the text encoder, it would be good to understand the effect of freezing vs. training it.

Benchmarking. For robustness to corruptions, standard evaluation is ImageNet‑C/P and modern robust fine‑tuning approaches (e.g., WiSE‑FT, which is cited in the related works), but none are evaluated in Tables 1/2. For dense localization, DenseCLIP would also be a suitable baseline. Without these, the 9% improvement headline is hard to contextualize. In general, the benchmarks in Tables 1 and 2 do not seem thorough enough.

Overclaiming from qualitative plots. The narrative that the decoder’s quality “directly causes” the feature‑space refinement is asserted using minimal qualitative inspection (e.g., in Figure 4, first row may simply be reflecting foreground-background contrast; second row focuses on the subject only despite uniform noise corruption). Stronger evidence is necessary to support this claim.

[1]: Masked Autoencoders Are Scalable Vision Learners (https://arxiv.org/abs/2111.06377)

**Questions:**

1. What exactly is the primary task? Is your main claim about robust classification, corruption detection, or localization?
2. Why are FLYP and “Ours (w/o Tune)” identical in Table 1? Is “w/o Tune” conceptually the same as FLYP? If not, what is the difference?
3. Benchmark choices. Given the claims about robustness to corruptions, why not report ImageNet‑C/P?
4. Choice of hierarchy layers (4/8/12). Why these layers specifically? What would happen if different/more layers were used?

---

### Official Review · Reviewer_D7Ut · 2025-11-06

**Soundness:** 3
**Presentation:** 3
**Contribution:** 2
**Rating:** 6
**Confidence:** 4

**Summary:**

This paper introduces CG-CLIP, an approach designed to enhance the OOD robustness of vision-language models like CLIP. The authors hypothesize that CLIP's inherent object-centric bias impairs its corruption robustness and generalizability. To counteract this, they propose integrating an auxiliary task, predicting pixel-wise corruption maps. The core of their method is a three-stage training curriculum ("Adapt-Isolate-Tune") designed to stably integrate this dense prediction task without disrupting the encoder's powerful pre-trained features. Experimental results demonstrate that CG-CLIP achieves significant performance improvements on corruption detection benchmarks for both in-domain and out-of-distribution data.

**Strengths:**

1. The paper tackles the practical issue of poor robustness to common image corruptions in large vision language models. Improving the reliability of these widely used models is a valuable and timely research direction.

2. The method's design is highly practical, as the auxiliary decoder is discarded after training. This ensures the final model gains robustness without any additional computational cost at inference time, a highly desirable property for real-world deployment.

3. The paper is clearly written, and the authors effectively demonstrate their methodology. The "Adapt-Isolate-Tune" curriculum is explained with sufficient detail and helpful visualizations, which aids in understanding and reproducibility.

**Weaknesses:**

1. The paper proposes multi-stage process is efficient but lacks sufficient justification over simpler self-supervised learning alternatives. For instance, a single training run where the auxiliary loss weight λ is gradually annealed from 0. Without this comparison, the necessity of the rigid and complex "Adapt-Isolate-Tune" pipeline is not convincingly demonstrated.

2. The "Isolate" stage is justified as a way to calibrate the decoder's gradient signal. How sensitive is the final performance to the duration of this stage? Is there a minimum number of epochs required for this calibration, and does prolonged training in this stage offer diminishing returns or even harm the final co-adaptation?

3. The corruption map generation relies on a frozen VGG-16, effectively using its perceptual feature space as a form of supervision. How sensitive are the final results to this specific choice? Would the performance gains hold if a different feature extractor were used to generate the maps?

**Questions:**

see Weaknesses

---

### Note · Authors · 2025-11-12

**Comment:**

I would like to formally withdraw my submission from ICLR. I sincerely thank all the reviewers for their time and constructive feedback.

**Withdrawal Confirmation:**

I have read and agree with the venue's withdrawal policy on behalf of myself and my co-authors.